# Experiences of Pediatric Pain Professionals Providing Care during the COVID-19 Pandemic: A Qualitative Study

**DOI:** 10.3390/children9020230

**Published:** 2022-02-09

**Authors:** Tieghan Killackey, Krista Baerg, Bruce Dick, Christine Lamontagne, Raju Poolacherla, G. Allen Finley, Melanie Noel, Kathryn A. Birnie, Manon Choinière, M. Gabrielle Pagé, Lise Dassieu, Anaïs Lacasse, Chitra Lalloo, Patricia Poulin, Samina Ali, Marco Battaglia, Fiona Campbell, Lauren Harris, Vina Mohabir, Fareha Nishat, Myles Benayon, Isabel Jordan, Jennifer Stinson

**Affiliations:** 1Child Health Evaluative Sciences, The Hospital for Sick Children, Toronto, ON M5G 0A4, Canada; tieghan.killackey@sickkids.ca (T.K.); chitra.lalloo@sickkids.ca (C.L.); lauren.harris@sickkids.ca (L.H.); vina.mohabir@sickkids.ca (V.M.); fareha.nishat@sickkids.ca (F.N.); 2Department of Pediatrics, University of Saskatchewan, Saskatoon, SK S7N 0W8, Canada; dr.kbaerg@usask.ca; 3Department of Anesthesiology and Pain Medicine, University of Alberta, Edmonton, AB T6G 1C9, Canada; bruce.dick@ualberta.ca; 4Department of Anesthesiology and Pain Medicine, University of Ottawa, Children’s Hospital of Eastern Ontario, Ottawa, ON K1N 6N5, Canada; clamontagne@cheo.on.ca; 5Department of Anesthesia and Perioperative Medicine, Western University, London, ON N6A 3K7, Canada; raju.poolacherla@lhsc.on.ca; 6Department of Anesthesia, Pain Management & Perioperative Medicine, Dalhousie University, Halifax, NS B3H 4R2, Canada; allen.finley@dal.ca; 7Department of Psychology, University of Calgary, Calgary, AB T2N 1N4, Canada; melanie.noel@ucalgary.ca (M.N.); kathryn.birnie@ucalgary.ca (K.A.B.); 8Department of Anesthesiology, Perioperative and Pain Medicine, Alberta Children’s Hospital Research Institute, University of Calgary, Calgary, AB T2N 1N4, Canada; 9Department of Anesthesiology and Pain Medicine, Faculty of Medicine, Research Center of the Centre Hospitalier de l’Université de Montréal, Université de Montréal, Montreal, QC H2X 0A9, Canada; manon.choiniere@umontreal.ca (M.C.); gabrielle.page@umontreal.ca (M.G.P.); 10Department of Biomedical Sciences, Research Center of the Centre Hospitalier de l’Université de Montréal, Université de Montréal, Montreal, QC H3C 3J7, Canada; lise.dassieu@umontreal.ca; 11Department of Health Sciences, Université du Québec en Abitibi-Témiscamingue, Rouyn-Noranda, QC J9X 5E4, Canada; anais.lacasse@uqat.ca; 12Department of Anesthesiology and Pain Medicine, University of Ottawa, Otttawa, ON K1N 6N5, Canada; ppoulin@toh.ca; 13Department of Psychology, The Ottawa Hospital Research Institute, Ottawa, ON K1Y 4E9, Canada; 14Departments of Pediatrics & Emergency Medicine, Faculty of Medicine & Dentistry, University of Alberta, Edmonton, AB T6G 2R3, Canada; sali@ualberta.ca; 15Department of Psychiatry, University of Toronto, Toronto, ON M6G 1H4, Canada; marco.battaglia@camh.ca; 16Division of Child and Youth Psychiatry, CAMH, Toronto, ON M6J 1H4, Canada; 17Department of Anesthesia and Pain Medicine, Hospital for Sick Children, Toronto, ON M5G 1E2, Canada; fiona.campbell@sickkids.ca; 18Department of Medicine, McMaster University, Hamilton, ON L8S 3L8, Canada; myles.benayon@gmail.com; 19Independent Researcher, Squamish, BC, Canada; isabeljordan@me.com; 20Lawrence S. Bloomberg Faculty of Nursing, University of Toronto, Toronto, ON M5T 1P8, Canada

**Keywords:** COVID-19, pediatric pain, pain clinics, distance treatments, e-health, telehealth

## Abstract

Chronic pain affects 1 in 5 youth, many of whom manage their pain using a biopsychosocial approach. The COVID-19 pandemic has impacted the way that healthcare is delivered. As part of a larger program of research, this study aimed to understand the impact of the pandemic on pediatric chronic pain care delivery including impact on patients’ outcomes, from the perspective of pediatric healthcare providers. A qualitative descriptive study design was used and 21 healthcare providers from various professional roles, clinical settings, and geographic locations across Canada were interviewed. Using a reflexive thematic analysis approach 3 themes were developed: (1) duality of pandemic impact on youth with chronic pain (i.e., how the pandemic influenced self-management while also exacerbating existing socioeconomic inequalities); (2) changes to the healthcare system and clinical practices (i.e., triaging and access to care); (3) shift to virtual care (i.e., role of institutions and hybrid models of care). These findings outline provider perspectives on the positive and negative impacts of the pandemic on youth with chronic pain and highlight the role of socioeconomic status and access to care in relation to chronic pain management during the pandemic in a high-income country with a publicly funded healthcare system.

## 1. Introduction

Chronic pain affects approximately 1 in 5 youth [1], impacting all domains of life including academic [2,3,4], psychosocial [5], and physical functioning [6]. Due to the complex nature of chronic pain, a biopsychosocial model of care incorporating psychological, physical, and pharmacological approaches is widely accepted as best practice [7,8,9,10]. Youth may be referred to a pediatric multidisciplinary chronic pain clinic (PMCPC) or an intensive rehabilitation program when community level interventions are not enough to achieve optimal pain management and improved function. These multidisciplinary programs commonly have a team which includes a medical doctor (e.g., pain physician, anaesthesiologist, or paediatrician), physical therapist, psychologist and/or psychiatrist, and a registered nurse or nurse practitioner. They may also include occupational therapy, social work, and/or child life specialties. Key components include joint goal setting with emphasis on functional rehabilitation (e.g., school attendance, fitness goals, stress management) and active strategies to support pain self-management [11]. 

Beginning in March 2020, in response to the COVID-19 pandemic, Canadian public health restrictions led to an abrupt pause of in-person pain care services at the community and tertiary care levels [12]. This limiting of in-person access to treatment lead to extensive consequences for people with chronic pain, both at the individual (i.e., worsening of condition) and the public health level (i.e., prevention of chronic pain) [12]. In order to address this concern, all tertiary-level Canadian pediatric pain clinics transitioned to virtual care (telephone and video-based service delivery) in response to widespread pandemic restrictions on in-person service delivery [13]. 

A recent survey by the study authors explored the impact of the rapid shift to virtual care during the COVID-19 pandemic on Canadian pediatric chronic pain clinics and healthcare providers (HCP), including the challenges that accompanied this shift [14]. This study was part of a larger multi-phase project supported by Canadian federal funding (Canadian Institutes of Health Research). Study results indicated that technological issues (i.e., Internet connection, access to technology, technological literacy) were the most common challenges to delivering virtual care, followed by administrative, financial, and logistical barriers related to infrastructure and setting up virtual care capabilities [14]. The present study builds on that research by generating in-depth qualitative insight into the experiences of HCP treating youth with chronic pain during the COVID-19 pandemic across community and tertiary care settings. We also aimed to explore the perspectives of HCP regarding their observations on the impact of the COVID-19 pandemic on youth with pain. 

## 2. Methods

### 2.1. Participants & Design

This study employed a descriptive qualitative design, as outlined by Sandelowski [15], to describe the experiences of HCP providing pediatric chronic pain care in the context of the COVID-19 pandemic. Utilizing purposive sampling with a focus on maximum variation, HCP from across care settings in Canada were recruited, representing a variety of professional roles, clinical settings, and geographical locations. Canada’s public healthcare system provides primary (e.g., family physician) and tertiary (e.g., PMCPC) care, however, many specific treatment modalities (e.g., physiotherapy) are not publicly covered, but may be paid for out-of-pocket or with private insurance. Eligible HCP were required to provide chronic pain care during the pandemic to youth under the age of 18 years and to be able to speak and read English. 

The institutional research ethics board of the Hospital of Sick Children approved the study (REB #1000070100). Eligible HCP were identified from a list of those who had opted-in for an interview following the completion of a national online survey as part of a wider program of research. Originally, the survey was widely disseminated to reach eligible HCP through medical leads of each PMCPC in Canada, as well as through the communication channels of several national HCP groups, including Canadian Pain Network, College of Family Physicians of Canada (Child and Adolescent Health Interest group), and Pediatric Emergency Research Canada. Research staff contacted eligible HCP via email to provide more information about the study, invite participation, and obtain informed consent. HCP who agreed to participate consented using an online consent form through Research Electronic Data Capture (REDCap), a secure online data collection tool [16]. 

Interviews occurred within the second wave of the pandemic in Canada, between the months of August and November 2020. All of the interviews were virtually conducted by two female team members [TK and VM]; the lead author (TK) is a postdoctoral fellow with doctoral-level training in advanced qualitative methodology, and VM is a clinical research project assistant who was trained in qualitative interviewing and supervised by the lead author. The interviewers had no previously established relationship with participants prior to the study’s commencement and the participants learned about the research and the background of the interviewers at the outset of the consent and interview process. Interviews were conducted using Zoom for Healthcare and were audio-recorded for verbatim transcription and analysis. A semi-structured interview guide was developed based on previous Severe Acute Respiratory Syndrome (SARS) impact work [17] and parallel qualitative work being completed with HCP who care for adults with chronic pain [8,10] (See Appendix A for the HCP Interview Guide). The guide focused on exploring HCP experiences working within the context of the COVID-19 pandemic, the impact on their patients, as well as the impact on their work and clinical care routines. Interviewers used prompts to elicit detail and clarity from participants regarding responses to interview questions (e.g., “What does that mean to you?” “Tell me more about that”) and were modified in an iterative fashion to incorporate specific questions based on analysis of preliminary interviews. Interviews were conducted at one time point and ranged in length between 31 and 68 min (mean duration = 44 min). HCP were each provided a $30 CAD gift card honorarium for their participation. 

### 2.2. Data Analysis

Data analysis was guided by a reflexive thematic analysis approach as outlined by Braun and Clarke[18,19]. Transcripts from the interviews were uploaded and analyzed using Dedoose software (Dedoose Version 8.3.47, https://www.dedoose.com/ accessed on 16 February 2021) [20], a cross-platform cloud-based application for analyzing qualitative and mixed methods data. 

In the early stages of data collection, two authors (TK, VM) met weekly to reflexively discuss the process of data collection and emergent patterns of early analysis. Reflexivity was applied both individually and as a team by discussing their relationship to and interest in the research topic, as well as considering how their academic, professional, and personal backgrounds influenced the generation and interpretation of data throughout the analytic process. As data collection and preliminary analysis progressed, both team members became familiar with the data by reading full transcripts and outlining key concepts for the codebook development process. Initial analyses to generate codes were conducted by the first author (TK) after iterative review of a first set of transcripts. After the initial codebook was established, all three team members (TK, VM, FN) read and coded the first four transcripts. Codes were compared, collapsed, or expanded, and the codebook was iteratively restructured several times to account for new data as transcripts were coded. After coding 18 transcripts, the codebook was finalized and there were no new codes being developed. Once all of the transcripts had been coded once, two team members [VM, FN] then reviewed all of the previously coded transcripts with the finalized codebook to ensure all codes were captured. Upon completion of coding, potential themes were constructed through a process of quotation review to develop an understanding of the “essence” of each code [19]. A team of coauthors [TK, VM, FN, LH, JS] reviewed the themes and contributed to the development of theme names and descriptions. Themes were finalized through the process of writing up results and weaving participant data (quotations) into an analytic narrative [19]. Quotations from a range of participating HCP were included to ensure a variety of perspectives are represented across each theme [19].

Reflexivity was incorporated throughout the research process and specifically into the interviews and analysis to support the rigor of this study [21]. Data generation and analysis was conducted by the first author [TK] and two research staff [VM, FN], all of whom have an interest in chronic illness management and come from a range of backgrounds (i.e., clinical, research, and lived experience of chronic pain). 

## 3. Results

A total of 21 HCP were invited to be interviewed from across Canada, from a group of *n* = 40 eligible. Participants were invited based on maximum variation of gender, province, role, and clinical context, with the largest cohort of providers from Ontario (*n* = 10/21, 47.6%) (see Table 1). The majority of HCP identified as female (*n* = 16/21, 76.2%), were physicians (*n* = 12/21, 57.2%) and worked in tertiary care (*n* = 11/21, 52.4%). Providers ranged in experience from less than 5 years (*n* = 5/21, 23.8%) to over 15 years (*n* = 4/21, 19.0%). 

Qualitative results have been organized into three broad overarching themes that each describes a unique aspect of the experience of HCP and the impact of the pandemic. Specifically: (1) duality of pandemic impact on youth with chronic pain; (2) changes to the healthcare system and clinical practices; (3) the shift to virtual care. These themes and their corresponding sub-themes are presented below. 


*Theme 1: Duality of Impact for Youth with Chronic Pain*


This first theme centers around the pandemic’s dual impact on youth living with chronic pain, which simultaneously triggered both improvement and deterioration for patients. HCP reported that their patients’ experiences varied widely; some experienced improvements in pain during COVID-19 (i.e., due to fewer activities, less social anxiety related to attending school in-person, more sleep, etc.) and some spiraled into pain crises due to challenges such as isolation, routine change (i.e., lack of access to exercise, team sports, therapies), and increased anxiety from COVID:

“Many of our patients were doing much better without a lot of their typical stressors in their lives. Then of course we had the alternative cohort that was doing so much worse…it was a huge spectrum in terms of how the patients responded and how the families responded because for some of them it felt like they got this window of time to be able to work on this [pain management] and focus on this, and for some of them it just exacerbated everything. So it really did go both ways.”[HCP 001, Physiotherapist, PMCPC]

Similarly, another provider reported this polarity of experiences: 

“It worked in two directions. There were groups of our kids who became more symptomatic, who became anxious, who became withdrawn, their mental health did suffer, they were alone, there was uncertainty, they weren’t able to go to their providers from whom they usually got care. So I think that group of kids did suffer. But then there’s this other part which is that many of our kids who were experiencing anxiety and it was being compounded by school actually did very well, like they stopped fussing over school, they could go and do some online learning, and they were all reporting that they were actually sleeping better, they were more active than they were, they were happier.”[HCP 006, Physician, PMCPC]

According to HCP, this duality was characterized by the youth with chronic pain experiencing both positive and negative outcomes related to the pandemic, especially regarding the shift to virtual school along fewer social expectations and extra-curricular activities. Furthermore, HCP reported that some youth experienced improvements in sleep which positively impacted their pain and mood and may have had more opportunities to *“get out and get some fresh air and go for a walk”* [HCP 004, Physician, Community-Based Practice]. According to the HCP, these factors all contributed to improvements in both pain levels and mood (i.e., *“they were happier”* [HCP 006, Physician, PMCPC]). Conversely, there were also several significant harms related to the pandemic for youth living with chronic pain that were identified by the HCP. Both positive and negative impacts for youth living with chronic pain will be further examined through two key subthemes: (i) mental health and stress management, and (ii) COVID exacerbating existing socio-economic inequities. 

### 3.1. Mental Health and Stress Management

In addition to the impact that the COVID-19 pandemic had on patients’ pain experiences and on pain management services, HCP voiced concerns regarding the impact of the pandemic on youth’s mental health specifically. Providers noticed that anxiety and stress related to contracting the virus itself, along with the measures to control the virus (i.e., wearing personal protective equipment [PPE]) were especially prevalent in their pediatric patients: 

“Definitely some of them are nervous about coming into the hospital for in-person assessment, so we usually reassure them that we’re doing this safely that there’s PPE available. I think that [some patients] are more anxious about contracting this illness and then definitely we can feel that they are more anxious.”[HCP 010, Physician, PMCPC]

Beyond the stress and anxiety of the pandemic itself, HCP were especially concerned with the limitations in healthcare service access, and how that would impact their patients’ mental health, as many youth with chronic pain have co-occurring mental health concerns:

“I was concerned that my patients weren’t getting care and they weren’t getting access… they went a month or two with nothing… a lot of them were really spiraling. We started seeing self-harm cropping up, eating disorders popping up, increased suicidal ideation popping up.”[HCP 001, Physiotherapist, PMCPC]

The duality of limitations on activity was a complex phenomenon. HCP described how the experience of stress varied for youth with pain and observed that the pandemic impacted patients’ ability to engage in coping strategies that may have been previously effective for managing their chronic pain. Simultaneously, HCP explained that many patients’ mental health improved because activities were restricted during the pandemic:

“Some patients, because schools shut down, they felt less anxious in this way because they had less challenges. But you also could see the downside of that right… not getting challenged anymore, not getting outside, completely inactive, getting more back pain, being on the screens, having more sleeping difficulties, switching completely the sleep schedules, reverse sleep schedules, things like that.” [HCP 019, Physician, Community-Based Practice]

Many HCP noted an increase in youth mental health concerns that were often multi-factorial in nature but related to the combination of limited community services and stress of the pandemic: 

“I would say definitely the mental health [concern] has really increased, particularly pediatric mental health. Again, just the lack of outpatient resources and we certainly know that [with] chronic pain, obviously your pain is worse when your mental health is not as well and those two compound each other. I would see on an average shift now probably double the mental health cases that we had before, which is frustrating. The other night when I worked nights, more than 50% of my cases were mental health or social issues—the other one I’ve been seeing a lot of. So patients, where the tensions at home are so significantly high, that in some cases it’s actually unsafe for the children. I think there’s so much extra stress that people have not had to manage prior to this and there’s always something new.”[HCP 014, Physician, Pediatric Emergency Department]

HCP further expressed the significant role of parents, not only in chronic pain management but in supporting management of their children’s stress and anxiety induced or exacerbated by the pandemic: 

“It did have an impact on worsening mental health, so a lot of [our patients] had more anxiety, more depression, which was kind of interlinked with anxiety about the pandemic. Isolation at home. What’s gonna happen? Parents’ mental health affecting them. Even if the child wasn’t really particularly worried about it, they just saw their parents worried about it. That kind of impacted them.”[HCP 011, Clinical Psychologist, PMCPC]

HCP explained that having a child with chronic pain was already a stressor on parents and families, and that concerns related to COVID-19 increased stress levels and negatively impacted the mental health of families who had already had many responsibilities to juggle during this time. They also noted the gendered dimension of the pandemic impact:

“I think the ones that have young kids at home, they’re absolutely exhausted, particularly the mothers, because as usual, women are shouldering the brunt of this pandemic. Especially say you have a teenager with chronic pain now at home, and then you have a four and eight year old at home, right, you just can’t cope. Yeah they’re emotionally exhausted”. [HCP 003, Nurse, Tertiary Care (Other)]

Overall, the rise in mental health concerns for both pediatric chronic pain patients and their families was a pervasive theme throughout interviews with HCP regarding the harmful impact of the pandemic on youth. 

### 3.2. COVID-19 Exacerbating Existing Socio-Economic Inequalities

This second sub-theme examines the role of COVID-19 in relation to existing inequalities in Canadian society. For HCP, the duality of outcomes for youth was strongly related to socio-economic status (SES). HCP reported that those with higher SES traditionally managed to cope or *“do well”* [HCP 006] with chronic pain management, which continued into the pandemic, but those with lower SES experienced additional challenges related to the COVID-19 pandemic and management of chronic pain.

“Where they’ve got resources, they’ve done well. I think that our families that where it’s a combination of poverty, distance, education, poor mental health, have suffered. Now is that true pre-COVID? I’d say that combination of things has always conspired to make pain management very difficult.”[HCP 006, Physician, PMCPC]

As this provider noted, challenges related to SES were relevant considerations for families navigating pediatric chronic pain management prior to the pandemic; however, the challenges of the pandemic significantly exacerbated these concerns and inequities:

“We do have some families that are lower SES [socio-economic status], so in those families it was more striking for them. More quickly they had issues around getting food and in some cases, being able to pay their bills or things like that because some of them lost their jobs. For families that either had parents lose their jobs or have already difficulty or insecurity in terms of food or proper shelters, that [pandemic] really impacted them and impacted their health.”[HCP 011, Clinical Psychologist, PMCPC]

In addition to families who may have had to cope with new challenges related to lost income or unstable housing, providers also discussed observing a dichotomy in how some families with higher SES had access to additional healthcare, services, and technology, while families with lower SES were less able to afford or access additional supports, which impacted their coping and pain management by creating access barriers to healthcare. Beyond the ability to afford community services or childcare, one HCP highlighted several concerns that were exacerbated by the pandemic, specifically related to patients’ and families’ abilities to rapidly transition to virtual school and receive virtual healthcare. These concerns included having the appropriate technology to engage with virtual care, having previous experience and ability to “log on” to the virtual setting right away without delay, and the flexibility in schedule to be available for virtual appointments:

“I think it all boils down to socioeconomic status. Some families have less ability to afford help for health and generally have less access to services in the community. So the patients that have higher SES, their parents have the computers for virtual right away, less of a delay. Their parents are familiar with virtual, they logged on. Their parents have jobs that offices will allow them to work at home, so they were at home. And on the flip side the families with lower economic status where their parents might have a labour job or something and were not in the house or they didn’t have access to childcare, they didn’t have family members or hired help, they were often not available for appointments, hard to get a hold of them and help their child. They wouldn’t have time or energy; they would work all night and sleep all day. So it was really strikingly different, I am sure there are some in the middle.”[HCP 018, Physician, Intensive Rehabilitation Program]

Overall, providers expressed that many of these families likely *“needed supports before”* [HCP 012, Physician, PMCPC] and that the *“extra stress”* [HCP 014, Physician, PMCPC] of the pandemic exacerbated previously existing inequalities in the pediatric chronic pain care system.


*Theme 2: Impact of Changes to Healthcare System and Clinical Practices*


The second major theme explored the impact of changes to the healthcare system and healthcare routines on pediatric chronic pain care providers. Participants reported significant changes in their daily clinical routines that negatively impacted their ability to provide pain care across settings, whether that meant having to wear PPE in-person, being redeployed to new care settings, or limiting the patients they were able to see in clinic:

“When there was a response to the pandemic that first began in March [of 2020], our clinic was told no ambulatory patients and no non-urgent patients […] Clinic basically shut down for about two months and I was redeployed to inpatient for two months.” [HCP 001, Physiotherapist, PMCPC]

Two main factors were echoed by multiple participants across various disciplines. First, the immediate impact of the pandemic restricted their ability to see any ambulatory and non-urgent patients, and second, that HCP were often redeployed or unable to return to providing chronic pain care in a timely fashion. The two sub-themes related to this impact on the healthcare system are: (i) triaging of care, and (ii) access to services. 

### 3.3. Triaging of Care 

HCP reported that the pandemic significantly impacted their care routines, especially in the early stages when there were institutional requirements that patients be more strictly triaged based on type of care or service needed. Providers discussed that triaging was focused on *“urgent and emergent”* [HCP 008, Clinical Psychologist, PMCPC] care only; however, *“the definition of urgent and emergent was not clear”* [HCP 001, Physiotherapist, PMCPC]. HCP discussed how urgent or emergent *“was amazingly defined as life or death”* [HCP 001, Physiotherapist, PMCPC] and what this meant to them in the chronic pain context: 

“Initially our directive was that urgent and emergent meant that you would die in 7 days if you didn’t get seen. So that was the standard that truly at the beginning we could only see a patient if not seeing them put them at risk of dying within a week.” [HCP008, Clinical Psychologist, PMCPC]

HCP reported that they experienced a bias in their ability to be granted permission to continue to provide care because the term “chronic” was associated with their pain services: 

“I think our clinic was in a similar boat to [other pain clinics] but it felt like our clinic was one of the last to be granted permission to kind of resume because we have ‘chronic’ in our title not ‘acute.’”[HCP 001, Physiotherapist, PMCPC]

HCP also reported that many decisions were made from a biomedical perspective and lacked a focus on psychological or mental health, which is critical for patients living with chronic pain. Due to this biomedical focus, providers had to advocate to their institutions on behalf of their patients to be able to provide sufficient access, especially for mental health services: 

“Working in mental health, we see that if our patients are actively suicidal that would qualify. But in a tertiary care hospital setting the way of viewing that criterion was often based on more of a medical-physical view of urgency. But working in mental health there’s always that stress when you do have patients who have expressed suicidal ideations that actually if someone’s calling in saying they’ve become way more depressed, they’re really not functioning and they’ve lost their support system and they’re in a very challenging situation, there was some advocacy that had to be done and within the organization to say truly we need to reach out to those patients because that was a significant concern clinically”.[HCP 008, Clinical Psychologist, PMCPC]

Overall, this focus on urgent care meant that providers were often limited by strict definitions which ultimately lead to delays in care and significant impacts on services for youth living with chronic pain: 

“I think the piece that would have been more helpful for us is that idea of urgent care versus chronic care. We had the capacity to start a month before we did, and it was because of the wording of the hospital mandate or the Department of [Public] Health mandate, it was like you’re only allowed to do urgent care […]. So that was a little bit frustrating, that delay. And then on the other side I think that the hospital and the Department of [Public] Health wide “urgent care only” restriction put us back by a month.”[HCP 020, Clinical Psychologist, PMCPC]

Participants expressed significant concerns regarding the requirement to triage care to the most urgent cases as well as frustration regarding the variable definitions of what constituted an “urgent” case. Overall, this policy approach led to limited access to much needed services by the pediatric chronic pain population over the course of the pandemic. 

### 3.4. Access to Services 

The triaging of care described above impacted the access that individuals had to healthcare services. One participant described how this type of triaging led to some of their patients not receiving timely access to the treatments they needed to manage their pain: 

“Even though you were opening the building, we’re only allowing certain people to come in so we have to pick only the worst situation, the ones that were suffering the most, knowing that the ones that were not suffering the most probably in a couple of months would then be suffering the most. So we’re constantly catching up on people that are not getting timely access to medications and prescriptions. We are only dealing with the most emergent problem so it’s like always putting out fires but never getting to the point where we can manage. So it’s a big problem.”[HCP 018, Physician, Intensive Rehabilitation Program]

In addition to the need to triage pain care, HCP witnessed modifications or limitations to access of services in both the acute care and community care settings:

“We definitely had kids come into our service as a direct result of COVID-19 surgical delays. So delays in surgeries, I think [our hospital] cut surgeries by 70–80%, so we had kids who were not emergency surgeries, they were elective, not urgent, but required, and they developed pain as a result of waiting. So we have a complex care child that has hip dysplasia that had hip surgery, and he ended up going on narcotics, like waiting… crying up all night waiting for his surgery. So we’ve had a few of those referrals that we wouldn’t have otherwise gotten, that are developing pain as a result.” [HCP 003, Nurse, Tertiary Care (Other)]

This provider highlighted that restricting surgical care to only urgent cases resulted in children developing pain while waiting for surgery. Not only was it challenging to manage the unintended consequences of delays in care, but providers were also unable to create adequate discharge plans for patients without sufficient community services in place:

“What was so difficult and so frustrating was trying to come up with the safe discharge plan, and so there’s a few more cases where we actually ended up admitting a patient just because we literally could not get them what they needed successfully […] I just remember this child who is very spastic, and their home care almost stopped coming, they didn’t get physio, they didn’t get—it was enormous, it’s like respite care kind of dropped by the wayside. And the family brought him in and he was so much worse than he’s been a month before, just pain from spasticity and we actually did admit him to hospital because the parents were exhausted, they had no respite, they couldn’t get their regular home care, they couldn’t get the regular physio, but it was just so all-encompassing that in the end the best thing for the child was to bring him into hospital because that was the only way we could actually sort out some of the issues.”[HCP 014, Physician, Pediatric Emergency Department]

Beyond the acute care setting, clinicians broadly highlighted the role that community healthcare services play in the management of youth with chronic pain, and expressed their concerns regarding the widescale shutdown of many of these services:

“Children with pain are already socially isolated, they’re already vulnerable, and I was worried about the loss of supports that many of these kids really needed in the community. So even though we carried on with virtual visits, a lot of them, their community supports stopped. So they had home physio, they had community physio, or they had different supports through the school system at their school, counsellor at their school that literally stopped. So that was really difficult, and I was concerned about it.” [HCP 003, Nurse, Tertiary Care (Other)]

Youth living with chronic pain often rely on a range of services and receive multi-disciplinary care from a variety of providers in a variety of settings. The rapid shut-down of in-person health services across settings, along with the impact of losing school-based services were especially concerning for HCP:

“A lot of kids [with developmental disabilities] were receiving therapy at school. So they weren’t in school. They weren’t wearing braces, aren’t getting their normal contact with a therapist, [doing their] stretching, eating, routine.”[HCP 018, Physician, Intensive Rehabilitation Program]

The closure of healthcare and educational services during the pandemic also meant that HCP had to work closely with youth and families to adapt to shifting treatment plans and re-orient the goals of care to prevent regression for patients who had been participating in a functional rehabilitation model of care, as this provider noted: 

“One of my first concerns was that for kids that were on a fairly structured plan to return to function, and we were working closely with families and kids to increase their physical activity, school attendance, all those things and suddenly treatment stopped right? So we were trying to get them walking, whatever that treatment looked like, and actually for many of the kids their goals stopped too, they were working towards something, joining a sports team, being able to attend Comicon, all sorts of cool goals that the kids had that were suddenly off the table. And all sorts of treatments and interventions for going out, returning to function, that suddenly like very quickly changed. So one of my first concerns was about how do we maintain the status quo, you know not falling backwards in terms of their function but even just regrouping and hoping the families were able to maintain some semblance of routine, sleep, like all those things we work on were changing right?”[HCP 008, Clinical Psychologist, PMCPC]

Overall, the limited access to all types of rehabilitation services that accompanied the pandemic resulted in HCP witnessing a variety of negative impacts for youth managing their chronic pain, thus having to work creatively to ensure that youth were able to shift their goals and adapt their treatment plans. 


*Theme 3: Shift to Virtual Care*


This final theme captures the participants’ experiences with a relatively rapid, full-scale shift to the provision of virtual pain care, particularly from the perspective of those working in tertiary care and community-based practice:

“So then of course when the pandemic hit us […] we felt a little bit “oh god we have to make a sudden change”. So we were very assertive about that change and we sensed that obviously this [pandemic] is not going to be an issue that’s resolved so let’s just get on it and make it happen. So very quickly we basically flipped what we were doing in person—which was our usual kind of initial multi-d[isciplinary] assessments, interdisciplinary follow-ups, psychoeducational groups, different physio, OT, psychology, psychiatry sessions—into a virtual world.”[HCP 007, Nurse Practitioner, PMCPC]

Participants were particularly focused on three key sub-themes: (i) the role of institutions in guiding the transition to virtual care, (ii) the benefits and limitations of virtual care in pediatric chronic pain management, and (iii) the shift to a hybrid model of care. 

### 3.5. Role of Institutions

At the onset of the pandemic, participants described how in some cases, institutional leaders failed to make decisions quickly enough, leaving clinicians to take up administrative leadership roles on top of their changing clinical routines. 

“We were told we could not do any clinics at all. Our administration was not particularly helpful or supportive in setting up virtual access. They weren’t letting us use Zoom or Skype or anything like that initially. And weren’t interested in discussing it at that time. I guess it must have been about six weeks, finally started distributing a version of Zoom for Healthcare, which they felt was more secure. And allowed us to start booking initially some recheck appointments virtually.” [HCP 015, Physician, PMCPC]

While participants described some frustration with the lack of guidance from the institutions, some participants also noted that this lack of guidance enabled a more rapid shift to virtual care, one that bypassed the typical regulations or “red tape” that may have slowed down the process in the past: 

“Well something that would’ve taken five years for [the hospital] to do across the organization, happened in a week. So all of a sudden we just leveraged everything and forgot about the red tape you normally have to go to, you have to go through committees. […] The amount of red tape you have to go through to implement something is way too long sometimes, and so we proved that really it shouldn’t have to take five years. I think there was like a virtual committee looking at a five year plan and that got accelerated to a two day plan so that was awesome.” [HCP 003, Nurse, Tertiary Care (Other)]

HCP also reported some trial and error during the shift to virtual care trying multiple virtual platforms before landing on one that worked. Many participants described a type of bottom-up approach to shifting to virtual care, where the decisions were being made on the ground rather than by institutional guidance or practice standards. Overall, participants outlined how the extenuating circumstances of the pandemic created an environment that allowed for a more rapid shift to virtual care then would have been possible otherwise. 

### 3.6. Benefits and Limitations of Virtual Care

Participants noted both benefits and limitations inherent in the rapid shift to virtual care. Benefits largely focused on broader accessibility and convenience for patients and families, especially for those who live farther from the tertiary care centres. The convenience of having virtual appointments meant that parents did not need to take as much time off work, making the visit less stressful and less financially burdensome. One HCP described the benefit of now being able to include a second parent in virtual visits who previously may not have been able to participate in the child’s clinic appointment. HCP noted improved attendance and punctuality for appointments since the shift to virtual care:

“We figured out a way to do the psychoeducational groups [virtually] so that it provided more equality amongst our patient population geographically. Our interdisciplinary follow-ups could very well continue virtually if we wanted them to, so it provides us with much more flexibility and now that we’ve gotten it figured out its much more timesaving, it’s not as time consuming. In person it’s little things like having to check in at the desk, and park, and do all these things it takes time, people are late. Virtually we have found actually it worked out and with our patient population, we’ve noticed that there’s less cancellations, there’s less being late to the appointment because they can do it out of the comfort of their own home.”[HCP 007, Nurse Practitioner, PMCPC]

A specific benefit of virtually providing physical therapy was that it supports more active rather than passive modalities, and provides the youth with feedback on movement in their home environment:

“When you look at physio, via passive modality versus active right—like put an ultrasound on, TENS machine on or whatever versus give me some exercises and movement. And often times where people get stuck is when they do too much passive, not enough active. But if you’re doing virtual it’s almost all active. It may be because it forces you more into that more active thing and you are doing the exercises in your home environment. When you practice your exercises in the gym and in physio you’re like “am I doing it right at home?” Whereas if you’re doing it at home and practiced with somebody guiding probably going to be more likely to be just doing it at home because you practiced it.”[HCP 006, Physician, PMCPC]

Simultaneously, HCP noted a variety of limitations with the provision of virtual care. As previously mentioned, technological challenges were significant, particularly in those families of lower SES, where they may not have the equipment, adequate internet connection, or private space required for an effective virtual care visit:

“Families with very low levels of socioeconomic advantage or disadvantage typically live in small cramped quarters like trailers. And we often found that families would have to move outside, they would be subject to the rest of the family members coming and going, noise, no privacy, so we had poor connections, we had poor lighting, we had a lot of family members coming and going, lack of privacy, very difficult to do physical exams under those circumstances.” [HCP 006, Physician, PMCPC]

Technological literacy was also described as a barrier, particularly for conducting physical assessments: 

“We’ve done our best to try to do assessments virtually, but I found that the biggest challenge there actually is technology quality. So whether it is our end or their end, the cutting out and the freezing, very difficult. Second thing is the family on the other end’s awareness of how to use the technology appropriately basically having appropriate lighting and being able to position the screen in a way that works or even knowing where their camera is on their device. So you’re like show me both arms and you’re seeing [only one], and you’re like okay go down, over, in, out, show me both hands. They can’t do that.”[HCP 001, Physiotherapist, PMCPC]

Overall, HCP reported a variety of both positive and negative aspects of the transition to virtual care and highlighted several key considerations relevant to the continued use of virtual chronic pain care to youth specifically related to accessibility, privacy, and equity. 

### 3.7. Hybrid Model 

The concept of innovative virtual care which leverages both in-person and virtual modalities in a hybrid model was a theme that frequently arose throughout interviews, particularly when discussing the future of pediatric chronic pain care in Canada. Many HCP reported that they currently were or were planning to move towards a hybrid model of care, where initial intake visits are held in-person and routine follow-up appointments take place online, or where patients attend appointments in-person but some of the pain team members virtually attend the appointment. One HCP described how their hybrid model was established and has facilitated their ability to replicate their in-person multi-disciplinary clinic experience:

“So by the end of September, we will be at the point where we are literally replicating everything that we did physically, virtually, and we are now doing a hybrid. Our new patients they come in and get an in-person physical exam, but the other half the team is virtually chatting with them within the same room and the same appointment. And then we can help some patients come in for in-person physio. So we are basically replicating almost the exact same, as of the end of September, the same as what we did before the shutdown.” [HCP 007, Nurse Practitioner, PMCPC]

HCP described the potential benefit to applying a hybrid model in innovative ways; because the uptake of virtual care was so rapid and widespread, participants expressed the benefits of hybrid models becoming recognized as a viable model of care delivery:

“I would hazard a guess and say we’re never going to go back to primarily in-person appointments. I think virtual will continue to be used. And the extent of that, I’m not entirely sure, because there is something nice about having my in-person contact, but I think it’ll definitely as far as accessibility goes, improve that for a lot of youth in the province. As far as how that could change things for youth with chronic pain after the pandemic, I think it just gives them more tools, right, and more accessibility.”[HCP 021, Clinical Psychologist, PMCPC]

Overall, HCP reported a range of benefits to the use of virtual care and expressed that the pandemic allowed them the opportunity to explore innovative models of care which will impact the delivery of pain care for youth across the country moving forward.

## 4. Discussion

This qualitative study explored the experiences of pediatric chronic pain professionals providing pain care throughout the first two waves of the COVID-19 pandemic in Canada. Our analysis highlighted 3 main themes related to this experience from the perspective of HCP: (1) the duality of the impact of the COVID-19 pandemic restrictions on youth with chronic pain; (2) changes to the healthcare system and clinical practices; (3) the shift to virtual care among pediatric pain HCP. Similar findings have been reported by quantitative studies of HCP in other clinical areas, both adult and pediatric, across various regions of the world [7,8,12,17,22,23,24], however, this is one of the first qualitative studies to report on this phenomenon from the perspectives of HCP within the context of pediatric chronic pain care provision.

The first theme relates to the duality of impact of the COVID-19 pandemic on youth with chronic pain, where HCP witnessed a range of positive and negative outcomes for youth with chronic pain. For example, HCP observed that the normalization of virtual or alternative school learning environments allowed space and flexibility for youth with chronic pain to use pacing, rest, position changes and other positive pain coping strategies. While it was noted that many patients experienced improvements due to short term reduction in activities and related stress, avoidance of in-person schooling may not be uniformly beneficial for long-term outcomes.

A recent study also lead by a Canadian research team highlighted the presence of physical health concerns in youth as a risk factor for exacerbated mental health concerns in response to the pandemic, and that youth with pre-existing physical health conditions reported more impact on their mental health and physical health than those without [25]. This is consistent with the findings reported here, particularly the heightened anxiety and stress HCP reported seeing in their patients with pain related to contracting the virus and/or the related restrictions of the pandemic response.

A key finding within the first theme was that providers witnessed the heightened role of SES in mitigating or exacerbating the effects of the pandemic on youth with chronic pain. This finding aligns with previous work reporting a contrast in the care received by patients of varying SES during the pandemic [24]. In a German study, Vasconcelos et al. found that pediatric patients with chronic conditions and those who were socio-economically disadvantaged were more likely to be reported as receiving worse care during the pandemic, in comparison to those with minor concerns who were more likely to be reported as receiving better care [24]. A qualitative study of adults with chronic pain also reported that existing systemic inequities intermingled with the existence of chronic health conditions lead to worse health outcomes and more precarious financial positions [22] for adults of lower SES [26]. Moreover, it was found that most adult pain clinics reported worsening symptoms for adult chronic pain patients due to the reduction in complementary services and increased stress associated to the COVID-19 pandemic [26]. This duality of outcomes offers an opportunity to better understand the individual, institutional and societal mechanisms that can lead to better outcomes for this population and leverage the benefits that were observed during the pandemic and the associated rapid shift to virtual care.

The second theme describes pandemic-related changes to the healthcare system and clinical practices which impact access to care. Several studies also reported similar professional concerns from HCP such as difficulty planning discharge [11] or community support for their patients, while also noting changes to care-seeking behaviour as many pediatric patients with chronic conditions (i.e., pain or diabetes) were limiting contact with the healthcare system due to fear of exposure to COVID-19 [24,27,28]. Specific to pain professionals, several authors highlighted service disruptions [11,23,26] as a main point of concern; for example, the closure of physiotherapy and psychological services made chronic pain management more difficult, leading to the worsening of patients’ health conditions. Additionally, the fact that pain care was not being perceived as essential [29,30], the reduction in available staff through redeployment to pandemic-related services, and the extended closure of clinics limiting in-person or virtual care [26] has also been echoed among adult pain professionals [26]. This lack of access to pain care may have led to an increase in hospital admissions likely due to poorly controlled pain and/or deteriorating mental health. Indeed, higher levels of illness severity and acuity has been reported by HCP working in Canadian emergency departments during the first six months of the pandemic [14]. Moreover, these concerns are also reported among pain patients [8] who were increasingly relying on pharmacological pain relief as a consequence of limited access to physical and/or psychological pain management strategies during the pandemic.

The third theme developed from these interviews described the shift to virtual care, including the role of institutions, the benefits and limitations of virtual care, and trialling of hybrid models of care. This theme aligns with other studies which have noted the HCP-reported limitations of virtual care (e.g., inability to perform physical exams, missing non-verbal and physical cues) [9,31] and inequitable access to broadband Internet and technological devices [9,11,31,32] as major barriers to delivering virtual care. Furthermore, some HCP felt their institutions lacked information technologies and administrative support, electronic medical records system and the space and equipment to adequately implement virtual care [9,26,31,32]. A recent systematic review found that virtual care was under-utilized in pediatric chronic pain care, and outlined strategies for how and where virtual care can be better leveraged [33]. The gaps in current knowledge outlined in that review are consistent with the findings of this study, for example the need for a better understanding of the limitations of virtual care for all aspects of chronic pain care and identify potential impacts on therapeutic relationships.

This last theme also identified some administrative opportunities and highlighted the policy-level implications of these findings. Some participants indicated that the lack of bureaucratic or institutional-level red tape facilitated the rapid uptake of virtual care as a necessity, compared with the standard years-long approval process that was required prior to the pandemic. Moving forward it may be relevant for administrative leadership and policy makers to consider the benefits of this type of rapid uptake of technology to encourage flexible and responsive care delivery. This data also highlighted the fluid decision making around what constituted an “urgent” case and potential misconceptions about the importance of rapid intervention in chronic versus acute conditions, which may be relevant to consider in the development of triage protocols in the context of future pandemic planning and response initiatives.

This study expands the current literature by developing a deeper understanding of the experiences HCP faced while providing pediatric pain care during the pandemic. Despite the many challenges during the COVID-19 pandemic, these results offer an opportunity to recognize positive outcomes associated with the effects of pandemic-related restrictions, such as improved access to online school, flexible course delivery and increased emphasis on self-management. In addition, these interviews highlight professionals’ ability to rapidly pivot to virtual care in order to continue to deliver high quality integrated medical-behavioural healthcare [34] for youth with chronic pain, which has been documented by at least one of the tertiary care clinics [35]. There is also a critical need to understand the impact of the COVID-19 pandemic from the perspectives of youth with pain, and their family members to gain first-hand insight into the challenges and benefits that were observed and reported by HCP.

## 5. Strengths and Limitations

Some limitations should be considered while interpreting the findings from this study. First, this study used a maximum variation sampling strategy; however, all of the providers worked within the publicly funded Canadian healthcare system and therefore these results may not be transferable to pediatric chronic pain professionals in other countries that may have differently structured systems. Secondly, given the changing nature of the COVID-19 pandemic, interviews were limited to only one time-point and therefore these results are specific to a certain phase of the pandemic, specifically prior to widespread vaccination in Canada. Findings were not presented by region or province although public health measures varied by jurisdiction in Canada and therefore there were differences in how HCP experienced the pandemic across the county. Future qualitative studies may consider a longitudinal approach that could explore the changing perspectives of HCP as the pandemic evolves and public health restrictions fluctuate to adapt to infection rates. Thirdly, this study does not capture the perspectives of pediatric chronic pain patients or their families; however, this work is forthcoming as part of the larger program of research by our group.

A key strength of this study is in the diversity of HCP sample who took part in interviews, which includes physicians, nurses, psychologists, physiotherapists, and child life specialists in both community-based and hospital settings. This sample was diverse in both professional background and included pan-Canadian representation, allowing the authors to capture a wide variety of experiences. Furthermore, the study captured, in real-time, the experiences of pediatric pain professionals providing pain care during a major public health crisis. Finally, a rigorous and reflexive coding process was enacted whereby each transcript was coded and reviewed twice by different team members, which strengthens the quality of our results.

## 6. Conclusions and Future Directions

The COVID-19 pandemic impacted pediatric chronic pain care providers and care delivery in a variety of ways. These results highlight the challenges of continuing to provide pain care during a pandemic and outlines the potential opportunities in utilizing virtual care and developing hybrid models of care to increase accessibility of chronic pain care. These results also present an opportunity for a call to action for the pediatric chronic pain community. Despite care for patients with chronic pain often not being considered “urgent” by dominant definitions used in some health systems, the results from this study suggest that the lack of access to chronic pain care may have increased patient suffering, specifically from a mental health standpoint. This impact was reported to vary according to socio-economic status and exacerbate pre-existing social inequities. Overall, the decision to focus on traditionally or biomedically “urgent” cases during various stages of the pandemic may result in an increased burden on the healthcare system in the long term. In the future, hospital systems should aim to make more sensitive, data-driven decisions in the triaging of care, especially considering the ongoing and worsening nature of the COVID-19 pandemic.

Going forward, exploring the perspectives of youth with chronic pain and their families will be critical in better understanding how virtual learning and other impacts of the pandemic restrictions may have benefitted or hindered the health and wellbeing of this population and to examine how the benefits could be leveraged beyond the pandemic. Future research should focus on exploring these perspectives of youth living with chronic pain and their family members to understand their experience of managing chronic pain during the pandemic and determine evidence-based best practices to support both healthcare providers and youth with chronic pain through the pandemic and beyond.

## Figures and Tables

**Table 1 children-09-00230-t001:** Demographics.

**HCP Type**	***n* (%)**
Child Life Specialist	1 (4.8)
Clinical Psychologist	4 (19.0)
Nurse	2 (9.5)
Physician	12 (57.2)
Physiotherapist	2 (9.5)
**Years of work with pediatric chronic pain patients**	***n* (%)**
Less than 5 years	5 (23.8)
5–10 years	7 (33.3)
10–15 years	5 (23.8)
More than 15 years	4 (19.0)
**Work setting**	***n* (%)**
Community-Based Practice	3 (14.3)
Emergency Department	4 (19.0)
Tertiary care (Intensive rehabilitation program)	2 (9.5)
Tertiary care (Multidisciplinary chronic pain clinic)	11 (52.4)
Tertiary care (Other)	1 (4.8)
**Gender**	***n* (%)**
Female	16 (76.2)
Male	5 (23.8)
**Province or territory of work**	***n* (%)**
Alberta	2 (9.5)
British Columbia	3 (14.3)
Manitoba	1 (4.8)
Nova Scotia	2 (9.5)
Ontario	10 (47.6)
Quebec	2 (9.5)
Saskatchewan	1 (4.8)

## Data Availability

The data that support the findings of this study are available from the corresponding author, J.S. upon reasonable request.

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
