# Peer review of "Experiences of Pediatric Pain Professionals Providing Care during the COVID-19 Pandemic: A Qualitative Study"

_children, 2022, doi:10.3390/children9020230_

Round 1
Reviewer 1 Report
The paper is well written and will be of interest to the healthcare community. A larger sample size would have been nice. honorarium can cause conflict of interest. In this case it was small honorarium and probably necessary to complete the study.
Author Response
Response to Reviewer 1 Comments
Comment 1/1: The paper is well written and will be of interest to the healthcare community. A larger sample size would have been nice. honorarium can cause conflict of interest. In this case it was small honorarium and probably necessary to complete the study.
Response 1/1: Thank you for this review. The authors believe that this sample size was sufficient in order to achieve the goals of this qualitative descriptive study. The sample was achieved using a purposive recruitment and represented a heterogeneous sample in order to capture a variety of experiences. Interviews were conducted and analyzed until data saturation was achieved, that is to say until the point that no additional themes or data arose from new interviews. Regarding the honorarium, the authors believe that the amount provided was small enough to have mitigated any conflict of interest for the participants and was necessary in order to feasibly complete this study. This honorarium was also approved by the institutions research ethics board and deemed appropriate to compensate participants for their time.
Reviewer 2 Report
Manuscript title: Experiences of Pediatric Pain Professionals Providing Care during the COVID-19 Pandemic: A Qualitative Study
ID: Children-1546739
This is a timely manuscript investigating an important topic in the field of pediatric pain. Just in reviewing the impressive author list, it is evident that the opinions in the paper are well-represented by professionals from various disciplines, as well as a patient partner, across the country of Canada. The authors sought to examine changes in healthcare delivery and impact on youth with chronic pain secondary to the COVID-19 pandemic. Qualitative methods were well described and employed to examine responses to structured interviews with providers, identifying three major themes. Interview methods and questions were nicely presented in the Appendix. The authors include many direct responses from providers throughout the manuscript, which provides a rich and informative perspective on the impact of the pandemic on healthcare and patients. Discussion focused on both positive and negative impacts of the pandemic on pediatric pain patients’ care and wellbeing, with an important focus on socioeconomic inequities that were exacerbated during the pandemic, as well as thoughtful commentary regarding changes to healthcare delivery in general. Overall, the work adds a valuable and unique perspective to the literature, and is impressive given the short timeframe in which the study has been conducted and written up. Given the ongoing impact of the pandemic, the timing of dissemination of these findings could not be more important so pediatric pain providers around the world can take these factors into consideration and learn from them. It was a delight to read this very well written and conducted study.
A few minor revision points are suggested below by section.
Abstract
- The abstract talks about the aim of the study as understanding “the impact of the COVID-19 pandemic on pediatric chronic pain care delivery.” However, one of the three primary themes revealed by the qualitative analysis was about the impact of the pandemic on patients’ outcomes; the authors are encouraged to highlight that in the summary presented in the abstract as well (impact on patient outcomes is well described at the end of the introduction as a study aim).
Introduction:
- The final sentence of the first paragraph on page 1 (lines 88-89) contains the word “goal/s” twice; the second appearance at the end of the sentence prior to the parentheses could be deleted to read: Key components include joint goal setting with emphasis on functional rehabilitation.
Data Analysis:
- In the first sentence of page 4 (lines 157-161), the term “reflexive thematic analysis” is introduced, and a few sentences later it is also mentioned that authors “reflexively discussed” aspects of data analysis. Despite the reference provided to the type of analysis this is referring to, it isn’t entirely intuitive what is meant by the term “reflexive” in these two mentions. Could a brief definition be added to make this term and process more descriptive and relatable for the reader? This would benefit the understanding of the term when it is used later, such as in line 180 on the same page. The explanation in the last paragraph on page 4 (lines 184-187) of what reflexivity is in this context is excellent and much more descriptive, perhaps that could be moved further up in the page to when this term is first used.
Results:
- Is there any data to present on why 50% of the interview sample declined to participate? Are there any differences or unique aspects to the group who did participate (other than being primarily physicians, females, and working in a tertiary care setting) that should be reported and considered when interpreting the data?
- This is strictly a formatting comment, for some reason, the provider quote in the access to services section on page 10 is double spaced and italicized; this should be changed to match the format of the other provider quotes throughout the document.
Discussion:
- On page 15, the second paragraph beginning on line 652 discusses findings from another study regarding the impact of the pandemic on patients’ mental health; however, nothing from this current study is mentioned in this section, which is curious given that impact of the pandemic on mental health was one of the major findings from the first theme. The authors are encouraged to expand on this paragraph by directly relating their findings to the previous research that is mentioned. Then, the next paragraph beginning on line 656 can start “Another key finding…” which is a very nice paragraph that fully discusses the socioeconomic findings; the mental health finding deserves similar attention.
Conclusion and Future Directions:
- This suggestion is entirely up to the authors, as it speaks more to the desired impact of the paper, which is theirs to determine. The authors could use this section to make more of a “call to action” conclusion for health care systems to take provision of care for chronic pain populations more seriously. Specifically, they could argue that even though care for patients with chronic pain was not deemed “urgent” (by some of the definitions described), the results from this study strongly suggest that the lack of access to chronic pain care increased patient suffering (e.g., impact on mental health) differentially (e.g., socioeconomic status), which put more burden on the healthcare system in the long term. This could be an important conclusion that hospital systems could use to provide more sensitive, data-driven decisions for future triaging of care, which is unfortunately still necessary to the ongoing and worsening pandemic.
Author Response
Response to Reviewer 2 Comments
Comment 1/7: [Abstract] The abstract talks about the aim of the study as understanding “the impact of the COVID-19 pandemic on pediatric chronic pain care delivery.” However, one of the three primary themes revealed by the qualitative analysis was about the impact of the pandemic on patients’ outcomes; the authors are encouraged to highlight that in the summary presented in the abstract as well (impact on patient outcomes is well described at the end of the introduction as a study aim).
Response 1: Thank you for this note, the impact of the pandemic on patients’ outcomes has been added to the aim listed within the abstract.
Comment 2/7: [Introduction] The final sentence of the first paragraph on page 1 (lines 88-89) contains the word “goal/s” twice; the second appearance at the end of the sentence prior to the parentheses could be deleted to read: Key components include joint goal setting with emphasis on functional rehabilitation.
Response 2: This edit has been made.
Comment 3/7: [Data Analysis] In the first sentence of page 4 (lines 157-161), the term “reflexive thematic analysis” is introduced, and a few sentences later it is also mentioned that authors “reflexively discussed” aspects of data analysis. Despite the reference provided to the type of analysis this is referring to, it isn’t entirely intuitive what is meant by the term “reflexive” in these two mentions. Could a brief definition be added to make this term and process more descriptive and relatable for the reader? This would benefit the understanding of the term when it is used later, such as in line 180 on the same page. The explanation in the last paragraph on page 4 (lines 184-187) of what reflexivity is in this context is excellent and much more descriptive, perhaps that could be moved further up in the page to when this term is first used.
Response 3: As recommended, the description of reflexivity has been moved up to provide context to when the term is first used.
Comment 4/7: [Results]: Is there any data to present on why 50% of the interview sample declined to participate? Are there any differences or unique aspects to the group who did participate (other than being primarily physicians, females, and working in a tertiary care setting) that should be reported and considered when interpreting the data?
Response 4: The potential interview sample of n=40 refers to 40 healthcare providers who had opted-in to be invited to interview, and only n=21 of those were invited based on the goal of achieving maximum variation in terms of gender, province, role, and clinical context. Healthcare provider interviews were conducted until data saturation was achieved, therefore there was not a need to invite the additional n=19 eligible to interview. A minor edit was made (line 192) to clarify this point.
Comment 5/7 [Results]: This is strictly a formatting comment, for some reason, the provider quote in the access to services section on page 10 is double spaced and italicized; this should be changed to match the format of the other provider quotes throughout the document.
Response 5: This formatting has been fixed.
Comment 6/7 [Discussion]: On page 15, the second paragraph beginning on line 652 discusses findings from another study regarding the impact of the pandemic on patients’ mental health; however, nothing from this current study is mentioned in this section, which is curious given that impact of the pandemic on mental health was one of the major findings from the first theme. The authors are encouraged to expand on this paragraph by directly relating their findings to the previous research that is mentioned. Then, the next paragraph beginning on line 656 can start “Another key finding…” which is a very nice paragraph that fully discusses the socioeconomic findings; the mental health finding deserves similar attention.
Response 6: Thank you for this valuable comment - we have tied in our findings related to youth mental health to this paragraph.
Comment 7/7 [Conclusion and Future Directions]: This suggestion is entirely up to the authors, as it speaks more to the desired impact of the paper, which is theirs to determine. The authors could use this section to make more of a “call to action” conclusion for health care systems to take provision of care for chronic pain populations more seriously. Specifically, they could argue that even though care for patients with chronic pain was not deemed “urgent” (by some of the definitions described), the results from this study strongly suggest that the lack of access to chronic pain care increased patient suffering (e.g., impact on mental health) differentially (e.g., socioeconomic status), which put more burden on the healthcare system in the long term. This could be an important conclusion that hospital systems could use to provide more sensitive, data-driven decisions for future triaging of care, which is unfortunately still necessary to the ongoing and worsening pandemic.
Response 7: Thank you for this very insightful suggestion. We agree this would be an excellent addition to the paper, and have therefore added the following into the Conclusion section (lines 778-788):
These results also present an opportunity for a call to action for the pediatric chronic pain community. Despite care for patients with chronic pain often not being considered “urgent” by dominant definitions used in some health systems, the results from this study suggest that the lack of access to chronic pain care may have increased patient suffering, specifically from a mental health standpoint. This impact was reported to vary according to socio-economic status and exacerbate pre-existing social inequities. Overall, the decision to focus on traditionally or biomedically “urgent” cases during various stages of the pandemic may result in an increased burden on the healthcare system in the long term. In the future, hospital systems should aim to make more sensitive, data-driven decisions in the triaging of care, especially considering the ongoing and worsening nature of the COVID-19 pandemic.